# COVID-19 Vaccine Antibody Response in a Single-Center Urban Hemodialysis Unit

**DOI:** 10.3390/vaccines11071252

**Published:** 2023-07-18

**Authors:** Mingyue He, Rui Song, Zakir Shaik, Crystal A. Gadegbeku, Louise Enderle, Christina Petyo, Sally B. Quinn, Zoe Pfeffer, Kathleen Murphy, Steven Kelsen, Aaron D. Mishkin, Jean Lee, Avrum Gillespie

**Affiliations:** 1Lewis Katz School of Medicine, Temple University, Philadelphia, PA 19140, USA; 2Cleveland Clinic Glickman Urological, Kidney Institute, Cleveland, OH 44195, USA; 3DCI—Dialysis Clinic Inc., Philadelphia, PA 19129, USA

**Keywords:** COVID-19, mRNA vaccines, antibody response, hemodialysis, COVID-19 naïve, COVID-19 recovered

## Abstract

Background: The longitudinal response to the COVID-19 vaccines among patients on hemodialysis with and without prior SARS-CoV-2 infection has not been well characterized. Methods: To guide vaccination strategies in patients on hemodialysis, it is critical to characterize the longevity and efficacy of the vaccine; therefore, we conducted a prospective single-center monthly antibody surveillance study between March 2021 and March 2022 to investigate the dynamic humoral response to a series of COVID-19 mRNA vaccines in patients on hemodialysis with and without prior SARS-CoV-2 infection. Monthly quantitative antibody testing was performed using the Beckman Coulter Access SARS-CoV-2 IgG Antibody Test©, which detects IgG antibodies targeting the receptor binding domain (RBD) of the SARS-CoV-2 spike protein. Results: This cohort of 30 participants (mean age: 61 ± 3 years) predominantly self-identified as African American (97%) and male (53%). Eight participants (27%) had recovered from COVID-19 (recovered) before the vaccine initiation. All participants received two vaccine doses, and 86.6% received a 6-month booster dose. Among patients naïve to COVID-19, the antibody positivity rate (APR) was 55% post-first-dose, 91% post-second-dose, 50% pre-booster at 6 months, 100% post-booster, and 89% at 6 months post-booster. Recovered patients sustained a consistent 100% APR throughout the year. The naïve patients demonstrated lower peak antibody levels post-second-dose than the recovered patients (17.9 ± 3.2 vs. 44.7 ± 5.6, *p* < 0.001). The peak antibody levels post-booster showed no significant difference between both groups (27.1 ± 3.9 vs. 37.9 ± 8.2, *p* = 0.20). Two naïve patients contracted COVID-19 during the follow-up period. Conclusions: The patients naïve to COVID-19 exhibited an attenuated and foreshortened antibody response following two doses of the mRNA vaccines compared with the recovered patients, who maintained 100% APR before the booster dose. The 6-month booster dose counteracted declining immunity and stimulated antibody responses in the naïve patients, even in previously non-responsive patients. This observation implies that different booster vaccination strategies might be required for COVID-19-naïve and -recovered patients. Post-vaccination antibody testing may serve as a valuable tool for guiding vaccination strategies.

## 1. Introduction

The coronavirus disease 2019 (COVID-19) pandemic, caused by the SARS-CoV-2 virus, significantly increased the mortality and morbidity in patients with end-stage kidney disease (ESKD) undergoing hemodialysis [1,2,3,4]. In the early stages of the pandemic, patients receiving hemodialysis experienced a heightened mortality rate of 20–30% and a hospitalization rate of 50% [4]. Regularly attending shared dialysis centers and having compromised immune systems make patients on hemodialysis particularly susceptible to SARS-CoV-2 infections [5,6], making effective vaccinations an essential need.

In healthy individuals, COVID-19 vaccines elicit a robust immune response [7], yet their effectiveness seems diminished in individuals on hemodialysis [8,9,10,11,12]. The delayed and attenuated response to COVID-19 vaccines in this population is likely linked to impaired humoral response and improper B lineage memory formation caused by the ongoing uremic state and chronic inflammation [13,14]. These findings prompted policy adjustments advocating for a third vaccine dose. Most available data; however, focuses on the early immune response weeks after the initial two doses, leaving the longevity of the immune response after two doses and the third booster uncertain in patients on hemodialysis. Furthermore, mRNA vaccines represent a novel form of vaccination that utilizes messenger RNA (mRNA) which encode the viral antigen and rely on the hosts own cells to translate and elicit an immune response. The immunogenicity of the mRNA vaccine, as well as the durability of the generated immune response in patients receiving hemodialysis, warrant further investigation and monitoring.

Recent studies in the general population suggested that individuals who recovered from COVID-19 (recovered) exhibited different vaccination responses compared with those without a prior SARS-CoV-2 infection (naïve). Hall et al. reported that in a cohort of 35,768 healthcare workers, vaccine effectiveness decreased to 51% six months post-second-dose in COVID-19-naïve participants. This immunity conferred by previous infection waned after one year in unvaccinated individuals, while vaccine effectiveness consistently remained above 90% in individuals who were vaccinated following a SARS-CoV-2 infection [15]. Relevant data on the longitudinal dynamic humoral response to COVID-19 mRNA vaccines in patients on hemodialysis with or without a history of SARS-CoV-2 infection have not yet been fully described.

To inform optimal vaccination strategies for patients on hemodialysis, it is imperative to elucidate and characterize the duration and effectiveness of vaccine-induced immunity in both COVID-19-naïve and -recovered patients on hemodialysis. Our dialysis clinic was significantly impacted by the COVID-19 pandemic, with greater than 10% of the clinic patients dying from complications of SARS-CoV-2 infections between March and April 2020; thus, studying the effectiveness of the vaccine in our center was a research priority [16]. In this single-center cohort study of thirty patients that underwent maintenance hemodialysis, we described a year-long longitudinal antibody response to a succession of doses of mRNA vaccines in COVID-19-naïve and -recovered patients from March 2021 to March 2022. Based on data from the healthy population, we hypothesized that the immune response profiles of COVID-19-naïve and -recovered patients on hemodialysis would differ and COVID-19-recovered patients would exhibit a stronger antibody response and extend immunity following the initial two doses compared with the COVID-19-naïve patients.

## 2. Methods

### 2.1. Study Design

This was a non-interventional, observational, prospective, longitudinal, single-center cohort study. Participants were recruited from a Middle Atlantic hemodialysis facility. Adult patients (aged 18 years or older) that underwent in-center maintenance hemodialysis, with and without a history of SARS-CoV-2 infection, were eligible for this study. To assess the longitudinal antibody response, we performed monthly antibody surveillance testing from March 2021 to March 2022. This study was performed in compliance with the Declaration of Helsinki and received approval from the Temple Institutional Review Board. All participants provided their consent after being fully informed.

### 2.2. Study Participants and Data Collection

All study participants completed a brief questionnaire, self-identifying their gender, race, and any medical comorbidities. Adherence to the prevailing COVID-19 vaccination policy ensured that all participants received a minimum of two doses of COVID-19 mRNA vaccines. Monthly quantitative antibody testing was performed using the Beckman Coulter Access SARS-CoV-2 IgG Antibody Test^©^, which detects IgG antibodies targeting the receptor binding domain (RBD) of the SARS-CoV-2 coronavirus spike protein. Antibody-level test results were made available to patients and clinical staff upon request but were not employed for clinical decision-making purposes.

### 2.3. Explanatory Variables

For the statistical analysis, participants were stratified into two groups: the COVID-19-naïve group (with no self-reported or documented history of previous SARS-CoV-2 infection) and the COVID-19-recovered group (with a self-reported or documented history of prior SARS-CoV-2 infection before enrollment and having already recovered from a previous infection).

Antibody levels were reported as signal-to-cutoff (S/CO) ratios. A seropositive response was defined as an antibody titer level equal to or above 1.00 S/CO SARS-CoV-2 IgG, with any value below considered seronegative, according to the Beckman Coulter Access SARS-CoV-2 IgG Antibody Test Kit instructions. 

Patients who exhibited a persistently seronegative response following two vaccine doses were defined as non-responders, whereas those with a seropositive response after two doses were considered vaccine responders. The antibody positivity rate (APR) was defined as the percentage of individuals who show a seropositive response out of the total number of patients under consideration. 

The rapid attenuation response was defined as the transition from a seropositive to a seronegative response before the third dose in those who received it, or within six months of the first dose in those who did not. If the seropositive response persisted before the third dose, this was referred to as non-rapid attenuation.

The determination of antibody levels at different time points in relation to COVID-19 vaccination was as follows: Antibody levels after the first dose were determined using the antibody levels measured closest to and prior to the second dose. Antibody levels after the second dose were determined using the antibody peak levels following two vaccine doses, which referred to the highest antibody level measured using monthly quantitative tests before receiving the third dose. Antibody levels before the third dose were defined as antibody levels measured nearest to and prior to receiving the third dose. Antibody levels following the third dose were determined using the antibody peak levels after the third dose, which was defined as the highest antibody level measured using monthly quantitative tests after the third dose.

For the first 6 months, all participants underwent monthly surveillance using nasopharyngeal swab testing to detect SARS-CoV-2 infections. During the one-year follow-up, if patients exhibited symptoms suggestive of SARS-CoV-2 infection, they were tested. A positive nasopharyngeal swab RT-PCR result was interpreted as confirmation of a SARS-CoV-2 infection. 

### 2.4. Statistical Analysis

Analyses were performed using STATA 17. Categorical variables are reported as frequencies and percentages, while continuous variables are presented as means and standard deviations. Differences in categorical variables across groups were assessed using the chi-squared test as appropriate. Differences in continuous variables across groups were analyzed using *t*-tests as appropriate. The dynamics of continuous variables were examined using the paired *t*-test. Multi-logistic or linear regression models were employed to evaluate the relationship between the vaccination response and relevant risk factors as appropriate. A *p*-value of 0.05 or lower was regarded as statistically significant.

## 3. Results

### 3.1. Baseline Cohort Characteristics

Of the one hundred and forty patients approached in this Middle Atlantic urban hemodialysis facility, 30 patients volunteered to participate and to have their antibody levels checked monthly from March 2021 to March 2022. The self-reported baseline characteristics of all participants are summarized in Table 1 (additional variables are provided in Appendix A). The mean age of the participants was 61 ± 3 years, the majority were African American (97%) and male (53%), and the mean dialysis duration was 4.6 ± 0.7 years. Hypertension (60%) and diabetes mellitus (53%) were the most prevalent comorbidities. No immunosuppressive medication use was reported during the study.

The COVID-19-naïve group consisted of 22 patients (73%), while the COVID-19-recovered group included 8 patients (27%). Table 1 compares the baseline characteristics of the two groups, revealing no significant differences. 

### 3.2. Vaccination Status

In compliance with the Centers for Disease Control and Prevention (CDC) guidelines, during the study period, all the participants were administered two doses of one of the mRNA vaccines. Most participants (87%) received a third booster dose. Most participants (93%) received the BNT162b2 mRNA vaccine, while the remaining 7% received the mRNA-1273 vaccine. Patients who initially received the BNT162b2 mRNA vaccine were subsequently administered the BNT162b2 mRNA vaccine for their second and third doses. Similarly, those who were initially administered the mRNA-1273 vaccine also received the mRNA-1273 vaccine for their second and third doses. 

In the COVID-19-recovered group, the participants received their first dose of vaccine 131 ± 21 days after their previous SARS-CoV-2 infection. The average interval between the first and second doses was 32 ± 5 days, while the mean period between the second dose and the third booster dose was 6 ± 0.4 months. There were no significant differences in the dosing intervals between the COVID-19-naïve group and the COVID-19-recovered group (Table 1).

### 3.3. Antibody Response Following Two Doses of the Vaccine

The antibody positivity rate (APR) was defined as the percentage of individuals who showed a seropositive response out of the total number of patients under consideration. Among the 22 COVID-19-naïve patients in our study, 12 exhibited seropositivity after the first vaccine dose, resulting in an APR of 55%. This rate increased to 91% (20 out of 22) after the second dose. Subsequently, their antibody titers decreased over the following six months, leading to an APR of 50% (11/22) before the third dose. Two patients (9%) were non-responders and maintained a seronegative status until the third dose. A comparison between the responders and non-responders in the COVID-19-naïve group did not reveal any independent risk factors (Appendix A). In contrast, the APR in the COVID-19-recovered group was 100% (8/8) following the initial two doses and before the third dose, regardless of a decrease in antibody level.

The antibody levels in all participants peaked at 60 ± 6 days post-second-dose. The duration to reach peak levels after the second dose was not different between the COVID-19-naïve and -recovered groups (65 ± 8 days vs. 45 ± 3 days, *p* = 0.15). The peak levels of the COVID-19-naïve group following the second dose were statistically significantly lower than those of the COVID-19-recovered group, as illustrated in Figure 1 (17.9 ± 3.2 vs. 44.7 ± 5.6, *p* < 0.001).

### 3.4. Attenuated Antibody Response over Time

Both the COVID-19-naïve group and the COVID-19-recovered group demonstrated a significant decline in their antibody response when comparing the peak antibody levels post-second dose and the levels before the third booster dose. For the patients who did not receive the third dose, the antibody levels recorded approximately six months post-second dose were utilized for the analysis. Of the 22 COVID-19-naïve patients, the mean antibody level significantly decreased from 17.9 ± 3.2 post-second dose to 5.6 ± 2.4 pre-third-dose (delta 12.3 ± 2.3, *p* < 0.001 via a paired *t*-test). Among the eight COVID-19-recovered patients, the mean antibody level significantly decreased from 44.7 ± 5.6 post-second-dose to 22.7 ± 5.2 pre-third-dose (delta 22.0 ± 2.7, *p* < 0.001 via a paired *t*-test).

Prior to the twenty COVID-19-naïve responders receiving their third dose of COVID-19 vaccine, eleven (55%) maintained seropositivity, while nine underwent a rapid attenuation response, transitioning from a seropositive to seronegative status. No independent risk factors were identified when we compared responders who exhibited rapid attenuation responses with those maintaining seropositive status before the third dose in the COVID-19-naïve group (Table 2). In contrast, the COVID-19-recovered group sustained an APR of 100% (8/8) prior to the third dose, with no instances of rapid attenuation response observed.

### 3.5. Antibody Response Following Three Doses of Vaccine

In our study, all thirty participants were administered two doses of an mRNA vaccine, with a majority (26 out of 30) receiving a third booster dose. The following results are from the analyses involving post-third-dose data that excluded the four patients who did not receive this third dose. 

Of the COVID-19-naïve patients, 86% (19/22) received a third booster dose, and the APR for patients who received the third dose was 100% (19/19). The antibody level significantly increased from a mean of 5.0 ± 2.6 pre-third-dose to 27.1 ± 3.9 post-third-dose (delta 22.1 ± 4.2, *p* < 0.001 via a paired *t*-test; Figure 2). Notably, both non-responders underwent positive serologic conversion after the third dose, albeit with low levels, and subsequently reverted to seronegative status two to three months after the third dose. 

Antibody levels are reported as signal-to-cutoff (S/CO) ratios: mean ± standard deviation (SD); Delta = mean antibody level post the third dose—mean antibody level pre the third dose. (Left) The antibody level significantly increased from a mean of 5.0 ± 2.6 pre-third-dose to 27.1 ± 3.9 post-third-dose (delta 22.1 ± 4.2, *p* < 0.001, paired *t*-test) in COVID-19-naïve patients; (Right) The antibody level significantly increased from a mean of 23.8 ± 5.9 pre-third-dose to 37.9 ± 8.2 post-third-dose (delta 14.1 ± 2.5, *p* < 0.001, paired *t*-test) in COVID-19-recovered patients. The incremental rise in antibody levels did not significantly differ between the naïve and recovered groups (22.1 ± 4.2 vs. 14.1 ± 2.5, *p* = 0.28).

A total of 88% (7/8) of the COVID-19-recovered patients received a third booster dose, and the APR was 100% (7/7) following the third dose. The antibody level significantly increased from a mean of 23.8 ± 5.9 pre-third-dose to 37.9 ± 8.2 post-third-dose (delta 14.1 ± 2.5, *p* < 0.001 via paired *t*-test; Figure 2). The incremental rise in antibody levels did not significantly differ between the naïve and recovered groups (*p* = 0.28).

Among the participants who received the third dose, antibody levels peaked at 38 ± 4 days post-third-dose. There was no significant difference in the time interval to reach peak levels after the third dose between the COVID-19-naïve and -recovered groups (36 ± 4 days vs. 42 ± 13 days, respectively; *p* = 0.55). Furthermore, the peak antibody levels following the third dose did not show a significant difference between the COVID-19-naïve and -recovered groups (27.1 ± 3.9 vs. 37.9 ± 8.2, respectively; *p* = 0.20). 

### 3.6. Comparison of the Antibody Response Following Two Vaccine Doses with the Response Following the Third Booster Dose

The antibody response following three doses revealed two peaks, with one after the second dose and the other after the third dose. Figure 3 depicts the differences in these peaks between the two groups. Among the nineteen COVID-19-naïve patients who received the third dose, the peak antibody level post-third-dose was significantly higher than the peak level post-second-dose (27.1 ± 3.9 vs. 17.4 ± 3.7; delta 9.7 ± 3.5; *p* = 0.007; Figure 3). Conversely, among the seven COVID-19-recovered patients who received the third dose, the peak antibody level post-third-dose was lower, albeit not significantly different from the peak level post-second-dose (37.9 ± 8.2 vs. 44.1 ± 6.4; delta −6.2 ± 3.8; *p* = 0.07; Figure 3). There was a significant difference between the peak levels in the COVID-19-naïve group and the COVID-19-recovered group (9.7 ± 3.5 for the COVID-19-naïve group vs. −6.2 ± 3.8 for the COVID-19-recovered group, *p* = 0.02)

Antibody levels are reported as signal-to-cutoff (S/CO) ratios: mean ± standard deviation (SD); Delta = mean peak antibody level post the third dose—mean antibody level post the second dose. (Left) Among the nineteen COVID-19-naïve patients who received the third dose, the peak antibody level post-third-dose was significantly higher than the peak level post-second-dose (27.1 ± 3.9 vs. 17.4 ± 3.7; delta 9.7 ± 3.5; *p* = 0.007); (Right) Among the seven COVID-19-recovered patients who received the third dose, the peak antibody level post-third-dose was lower, albeit not significantly different from the peak level post-second-dose (37.9 ± 8.2 vs. 44.1 ± 6.4; delta −6.2 ± 3.8; *p* = 0.07). The difference between the peak levels in the COVID-19-naïve group and the COVID-19-recovered group was significant (9.7 ± 3.5 for the COVID-19-naïve group vs. −6.2 ± 3.8 for the COVID-19-recovered group, *p* = 0.02).

### 3.7. Six Months Post-Booster Antibody Response

Among the 19 COVID-19-naïve patients who received the booster, one contracted COVID-19 four months post-booster (presumed Omicron variant) and was excluded from this analysis. Of the remaining 18 patients, 16 tested seropositive, yielding an APR of 89% (16/18), with a mean antibody level of 12.9 ± 2.3.

Of the seven COVID-19-recovered patients who received the booster, antibody levels were tested for five individuals at the six-month mark; all were seropositive, giving an APR of 100% (5/5), with a mean antibody level of 30.2 ± 6.5. The antibody level of the COVID-19-naïve group was significantly lower than that of the COVID-19-recovered group six months after the three vaccine doses (12.9 ± 2.3 in naïve vs. 30.2 ± 6.5 in recovered; *p* = 0.005).

### 3.8. Overview of Longitudinal Trends in the Humoral Response to Vaccinations over Time

Figure 4 illustrates the longitudinal dynamic changes in the antibody response following a series of mRNA COVID-19 vaccinations in COVID-19-naïve and -recovered patients on hemodialysis. 

### 3.9. Incidents of COVID-19 Cases

During the first six months of surveillance testing for SARS-CoV-2 infections, no cases of asymptomatic SARS-CoV-2 infections were identified. Throughout the one-year follow-up period, two COVID-19-naïve patients contracted primary SARS-CoV-2 infections and required hospitalization. One patient became infected one month after receiving the first dose but before receiving the second dose. Another patient developed a SARS-CoV-2 infection four months after receiving the third dose. No cases of SARS-CoV-2 reinfections were identified in the COVID-19-recovered group.

## 4. Discussion

In this study, we characterized and compared the longitudinal antibody responses following a series of COVID-19 mRNA vaccine doses among patients that underwent hemodialysis, who were stratified by a previous SARS-CoV-2 infection as COVID-19-naïve and COVID-19-recovered. We found that the COVID-19-naïve and COVID-19-recovered patients on hemodialysis exhibited distinct antibody response patterns to the series of COVID-19 mRNA vaccines. COVID-19-naïve patients exhibited a delayed, attenuated, and foreshortened antibody response following two doses of an mRNA vaccine, with only half retaining seropositivity before the third dose (six months after the second dose). However, the third dose effectively counteracted the declining immunity and stimulated antibody responses, even in the previously non-responsive patients, thus conferring significant benefits for naïve patients. Conversely, the COVID-19-recovered patients displayed a more robust humoral response to the initial two doses of an mRNA vaccine and maintained seropositivity before the third dose, indicating a more enduring response to the mRNA COVID-19 vaccines. Thus, the COVID-19-recovered patients may have derived less benefit from the booster vaccination. Our study uniquely juxtaposed the vaccine responses of COVID-19-naïve patients and those who had recovered from prior SARS-CoV-2 infections. Earlier studies either excluded COVID-19-recovered patients or failed to differentiate between the two groups. Moreover, research that differentiated these two groups mainly focused on the immediate immune response weeks to months after the initial two doses. Our study provides a more comprehensive and longitudinal perspective, unveiling the significantly different trajectory of antibody response in COVID-19-naïve and -recovered patients in a one-year follow-up.

Our findings of a 55% APR post-first-dose and a 91% APR post-second-dose in the COVID-19-naïve patients on hemodialysis are similar to earlier research. Carr et al. pooled 22 studies of patients that received hemodialysis or peritoneal dialysis and observed seropositive rates of 18–53% after the first dose and 70–96% after the second dose [17]. These are modestly lower than the seroconversion rate observed in healthy controls (68% and 100%, respectively) [18]. These results unequivocally demonstrated that COVID-19-naïve patients undergoing hemodialysis mount a delayed and weaker immune response to vaccination compared with healthy controls; hence, the second dose is required and should not be delayed. The COVID-19 vaccination for patients undergoing hemodialysis is imperative because vaccine-induced immunity provides protection against COVID-19, which is associated with a reduced risk of SARS-CoV-2 infection, COVID-19-related hospitalization, and mortality in dialysis patients [19]. Furthermore, the mRNA-based COVID-19 vaccines have demonstrated substantial efficacy across diverse ethnic groups, and previous studies suggested elevated vaccine efficacy in the African American population [20]. Our cohort, which was predominantly composed of African American participants, manifested an antibody positivity rate that trended toward the higher end of previously recorded seropositivity rates, aligning with existing literature.

The COVID-19-recovered patients had an excellent APR of 100% after the first dose and remained at 100% throughout the follow-up period. Compared with the COVID-19-naïve patients, they had considerably higher antibody levels and longer persistent antibody responses. These findings demonstrated that two doses of vaccination following SARS-CoV-2 infection appeared to enhance a stronger antibody response, extend immunity, and provide longer protection for COVID-19-recovered patients. The underlying premise is that individuals who had contracted and subsequently recovered from COVID-19 had developed a degree of natural immunity to the virus. In this context, the administration of the initial dosages of the vaccine appeared to function in a manner akin to the booster doses, thus enhancing this pre-existing immunity. Interestingly, many studies have demonstrated that patients on dialysis have a robust and long-lasting humoral immune response to a natural SARS-CoV-2 infection, even one that is comparable to the response of the general population [21,22]. This may be attributed to survivor bias in light of the significant COVID-19-related mortality among patients on dialysis. However, immunity from prior infection does protect against reinfection, as shown in a study of 2337 dialysis patients. Immunity acquired from prior infections was associated with 45% and 79% reductions in the risks of COVID-19 reinfection and symptomatic reinfection, respectively [23].

Of note, our study incorporated six months of monthly SARS-CoV-2 infection surveillance testing, during which no asymptomatic infections were identified. This led to the decision to discontinue surveillance testing in the subsequent six months. This observation suggests that asymptomatic SARS-CoV-2 infection is relatively rare in patients undergoing hemodialysis, which is a finding that aligns with the results reported in other studies [24].

The antibody titers and immunity waned over time in both the COVID-19-naïve and -recovered groups. Almost half of the COVID-19 vaccine responders became seronegative six months after the initial two doses. Similarly, Anand et al. reported that in a dialysis cohort of 2563 patients with 69.7% COVID-19-naïve patients, the undetectable antibody response rate 5 to 6 months following vaccination was 31% for the BNT162b2 mRNA vaccine and 11% for the mRNA-1273 vaccine. Furthermore, 20.6% of the COVID-19-naïve patients were seronegative at 6 months, whereas 16.4% of the COVID-19-recovered patients were seronegative at 6 months, with 56 cases of breakthrough infections identified [25]. Persistent antibody responses were reported to be associated with higher initial titers, as seen in patients on peritoneal dialysis and those who were administered the mRNA-1273 vaccine [26,27]. These findings suggest that while two doses of the mRNA vaccine could provide considerable short-term protection to COVID-19-naïve patients, this protection significantly diminished after six months, necessitating a third booster immunization. In our study, the APR at six months (before the third dose) after the initial two vaccine doses was lower in the COVID-19-naïve group compared with previously reported results. This difference may be attributed to factors such as our cohort’s lower initial antibody titers, most participants receiving the BNT162b2 mRNA vaccine, and the small sample size. Because of our sample size, factors associated with differences in the durability of antibody responses, other than previous infections, remains unknown, underscoring the need for additional research. Conversely, the APR at six months post-initial-two-dose administration in the COVID-19-recovered group surpassed the previously reported results. This discrepancy was likely caused by higher initial antibody titers. Notably, in our study, COVID-19-recovered patients received their first dose of vaccine approximately 131 days following their previous SARS-CoV-2 infection. 

In our study, the third dose of the vaccine induced a weaker antibody response in the COVID-19-recovered patients than the second dose did in the COVID-19-naïve patients. Similar observations of a lack of significant improvement in antibody response in patients with already high humoral responses were described in other studies [12,28]. These findings imply that for the COVID-19-recovered patients, a third dose given within six months may be of limited benefit. These observations further propose that the scheduling of the third dose for patients on hemodialysis who have recovered from COVID-19 should diverge from that of COVID-19-naïve patients. Specifically, the interval between the administration of the second dose and the subsequent booster could be protractedly extended for COVID-19-recovered patients exhibiting elevated antibody levels.

For the COVID-19-naïve patients, the third booster dose administered six months after the second dose significantly boosted the antibody responses. This indicates that the third dose was highly effective at counteracting the waning immunity and the diminishing efficacy of the first two doses. Notably, the two non-responders who had remained persistently seronegative after the first two doses of vaccines exhibited positive antibody responses after receiving the third dose. Similar responses to the booster dose were observed in other studies. For example, Dekervel et al. reported that the antibody response was boosted by a third dose, with 54.5% and 95.2% of patients receiving dialysis transitioning from non-responder status to low-responder status and from low-responder status to high-responder status, respectively [29]. However, in our study, the antibody response of the two non-responders to the third dose was markedly foreshortened, lasting only two to three months. Therefore, the “non-responding” patients on hemodialysis may require a distinct vaccination strategy, such as higher vaccine doses or more frequent booster doses, to achieve long-lasting effective protection from vaccination. Early identification and detection of these “non-responding” patients is crucial. The small sample size prevented us from identifying any independent risk factors for the vaccine non-responders and this needs to be investigated in future studies with larger sample sizes.

Our study provides valuable insights that could shape vaccination strategies. For instance, we found that the antibody levels typically reached their peak approximately 60 days after the second dose and around 38 days following the booster dose. Therefore, it appears reasonable to perform antibody testing one to two months post-vaccination to assess a patient’s immune response to the vaccine and aid identification of non-responders/poor responders. Moreover, our observation of rapid antibody attenuation—as evidenced by some patients transitioning from a seropositive to seronegative status within six months post-vaccination—emphasizes the importance of follow-up antibody testing at the six-month mark post-vaccination. This measure could aid in identifying individuals experiencing rapid declines in their antibody levels, thereby suggesting the need for a booster shot. These findings underscore the utility of post-vaccination antibody testing as a strategic tool for optimizing individual vaccination approaches. This proposed strategy parallels the approach adopted for hepatitis B vaccination, wherein antibody testing is routinely recommended, and revaccination becomes warranted if anti-HBs levels drop below 10 mIU/mL following the primary vaccine series.

While the presence of antibodies signifies an immune response, it is crucial to acknowledge that this does not automatically equate to functional immunity or effective protection against SARS-CoV-2. Currently, in the absence of universally validated and accepted biomarkers for protection against SARS-CoV-2 infection, antibody testing can serve as a practical surrogate marker for devising optimal vaccination strategies in patients on hemodialysis [30]. The challenge of using antibody levels to guide vaccination strategies lies in determining protective antibody levels correlating with acceptable immunity. Recent research suggested thresholds for anti-spike antibodies at 264 BAUs/mL and anti-RBD antibodies at 506 BAUs/mL. These thresholds are notably associated with an 80% vaccine efficacy against symptomatic SARS-CoV-2 infection, primarily involving the Alpha variant. This efficacy level is generally deemed protective, reinforcing the relevance of these antibody levels as benchmarks in immunological assessments [31]. However, these thresholds require further exploration and validation in subsequent studies. Moreover, the protective efficacy of antibodies—whether engendered by prior infection or vaccination—against the continually emerging variants of concern (VOCs) in patients on hemodialysis remains a question for comprehensive exploration. The evolving nature of VOCs may call for revisions to antibody-level thresholds, adding another layer of complexity to this rapidly evolving field of study.

Our study must be interpreted within the context of its limitations. First, it was conducted in a single urban hemodialysis center and the limited sample size potentially constrained the statistical analysis, including the identification of independent risk factors linked to non-responsiveness or rapid attenuation. Moreover, our patient cohort was predominantly composed of African Americans (97%), which might limit the generalizability of our results. Nonetheless, given the paucity of research on vaccine efficacy and durability among African American patients undergoing hemodialysis, our study provides a unique viewpoint. Second, the assessment of the vaccination response was solely based on antibody levels, excluding considerations of cellular immunity. Despite this, previous research demonstrated a significant correlation between antibody response and cellular immunity [10]. Third, we cannot dismiss potential misclassification of prior infection status, as some patients lacked data on their antibody levels before the first dose, resulting in classification based on self-reported or documented SARS-CoV-2 infection history.

## 5. Conclusions

In conclusion, our research elucidates distinct trajectories of antibody responses to the initial two doses and subsequent booster of COVID-19 mRNA vaccines in COVID-19-naïve and -recovered patients on hemodialysis over a one-year follow-up. We observed a significant impact of prior SARS-CoV-2 infection on vaccination-induced antibody response in this cohort of hemodialysis patients. Our observations led us to propose novel recommendations that emphasize the importance of post-vaccination antibody testing. Specifically, this testing can facilitate the early identification of poor or non-responders, thereby enabling the timely implementation of revaccination strategies to improve humoral response. Moreover, for COVID-19-recovered patients exhibiting high antibody levels, a delay in administering the booster vaccination could be a viable option. Future vaccination strategies for patients on hemodialysis should be personalized by considering each patient’s unique vaccination-induced antibody response and their history of SARS-CoV-2 infections.

## Figures and Tables

**Figure 1 vaccines-11-01252-f001:**
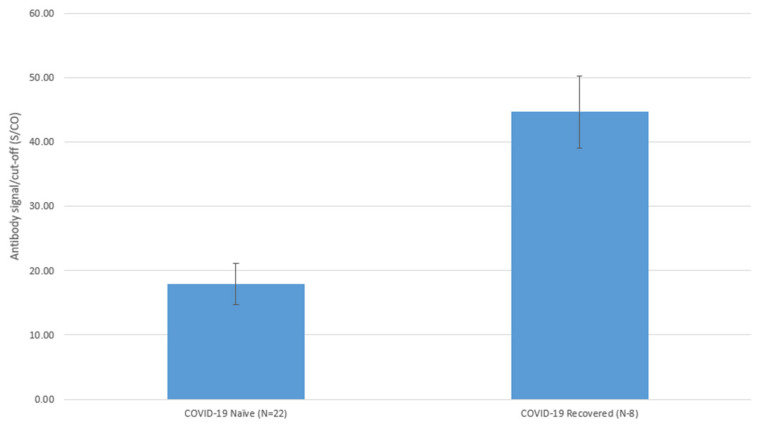
Differences in Antibody Peak Levels after Two Vaccine Doses in COVID-19 Naive and COVID-19 Recovered Patients on Hemodialysis. Antibody levels are reported as signal-to-cutoff (S/CO) ratios: mean ± standard deviation (SD); in COVID-19-naïve patients, the mean antibody level was 17.9 ± 3.2, while n COVID-19 recovered patients the mean antibody level was 44.7 ± 5.6, *p* < 0.001.

**Figure 2 vaccines-11-01252-f002:**
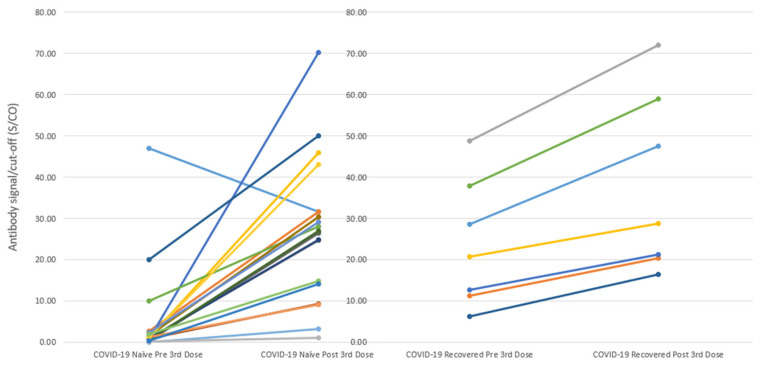
Changes of Antibody Levels from Pre and Post the Third Dose in COVID-19 Naïve and Recovered Patients on Hemodialysis.

**Figure 3 vaccines-11-01252-f003:**
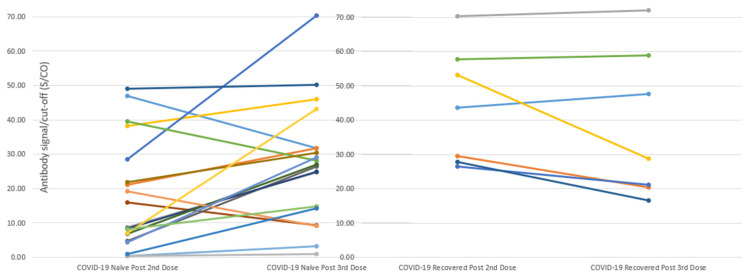
Changes of Peak Antibody Levels Post the Second Dose and Post the Third Dose in COVID-19-naïve and COVID-19-recovered Patients on Hemodialysis.

**Figure 4 vaccines-11-01252-f004:**
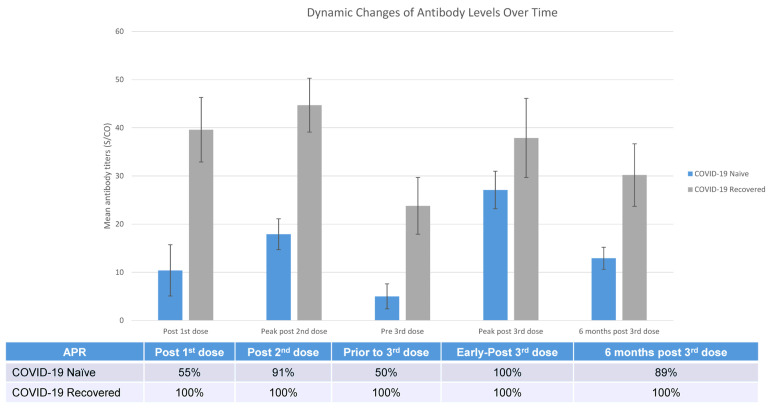
Overview of One-Year Longitudinal Dynamic Changes of Humoral Responses to a Series of COVID-19 mRNA Vaccinations in COVID-19-naïve and COVID-19-recovered Patients on Hemodialysis.

**Table 1 vaccines-11-01252-t001:** Baseline characteristics of the enrolled patients.

Characteristics	Total	COVID-19 Naïve N = 22	COVID-19 Recovered N = 8	*p*-Value
Age, years	61 ± 3	60 ± 3	63 ± 4	0.66
Female sex, no. (%)	14 (47%)	8 (36%)	6 (75%)	0.06
African American race, no. (%)	29 (97%)	21 (96%)	8 (100%)	0.54
Time on dialysis, years	4.6 ± 0.7	4.5 ± 0.9	4.9 ± 1.1	0.84
**Comorbidities**				
Hypertension, no. (%)	18 (60%)	13 (59%)	5 (63%)	0.87
Diabetes, no. (%)	16 (53%)	10 (46%)	6 (75%)	0.15
Obesity, no. (%)	1 (3%)	0 (0%)	1 (13%)	0.09
Chronic heart failure, no. (%)	3 (10%)	2 (9%)	1 (13%)	0.78
Cancer, no. (%)	3 (10%)	3 (14%)	0 (0%)	0.27
Active smoking, no. (%)	1 (3%)	1 (5%)	0 (0%)	0.54
**Types of vaccines**				0.44
BNT162b2mRNA	28 (93%)	21 (96%)	7 (88%)	
mRNA-1273	2 (7%)	1 (5%)	1 (13%)	
**Vaccine intervals**				
Between the first and second dose, days	32 ± 5	36 ± 7	23 ± 1	0.27
Between the second and third dose, months	6 ± 0.4	6 ± 0.4	7 ± 0.9	0.18

**Table 2 vaccines-11-01252-t002:** Adjusted logistic analysis comparing patients with and without rapid attenuation response in the COVID-19-naïve group.

Risk Factor		Adjusted OR	95% CI	*p*-Value
Age	Per year	0.95	0.83–1.08	0.42
Sex	Female	2.02	0.13–31.45	0.62
Time on dialysis	Per year	1.07	0.78–1.46	0.68
Hypertension	If yes	0.19	0.01–4.07	0.29
Diabetes	If yes	0.36	0.02–6.14	0.48
Interval between first and second dose	Per day	0.95	0.84–1.06	0.35
Peak antibody level after second dose	Per 1 S/CO	0.90	0.80–1.01	0.09

## Data Availability

All data used in this study are included in this article and its Appendix A.

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
