# Peer review of "COVID-19 Vaccine Antibody Response in a Single-Center Urban Hemodialysis Unit"

_vaccines, 2023, doi:10.3390/vaccines11071252_

Round 1

Reviewer 1 Report

The topic itself is important and welcome, however, there are some remarks to be made.

The biggest issue is the extremely low number of participants, meaning 22 and 8 patients per group. As far as I can see, this isn't even appropriately addressed in the discussion and methods sections. Even the references the authors use for similar studies included more than 2000 subjects. 

Also, 97% of the participants were African Americans. The potential impact of this on the results is not discussed. 

The authors repeatedly use the term "COVID-19 infection". This is inappropriate since COVID-19 is the disease itself, caused by an infection by Sars-CoV-2. Similarly, one wouldn't say "AIDS infection". It is either AIDS, the disease, or an infection caused by HIV, which aren't necessarily the same. 

Technically, there were two further subgroups, one receiving the Pfizer, and the other one receiving the BioNTech vaccine. We cannot presume that they had the exact same effect. 

How was the arbitrary cutoff value for "responders" and "nonresponders" determined? I see no reference for this. 

The English of the manuscript is acceptable. No major issues. 

Author Response

Dear Reviewer:

On behalf of all the contributing authors, we sincerely appreciate the opportunity to resubmit the revised draft of our manuscript. Your invaluable time and effort, along with the detailed feedback and constructive comments, have significantly contributed to enhancing the quality of our work. We are presenting our responses to your comments and concerns in a point-by-point manner.

Reviewer comments are presented below in italicized font, with individual concerns numbered for easy reference. Our responses follow in regular font. Changes and additions made to the manuscript based on these suggestions are indicated in red text.

Comments from Reviewer:

  1. The biggest issue is the extremely low number of participants, meaning 22 and 8 patients per group. As far as I can see, this isn’t even appropriately addressed in the discussion and methods sections. Even the references the authors use for similar studies included more than 2000 subjects.

Response: We appreciate your insightful feedback. Consequently, we have incorporated the following statement into the introduction: “Our dialysis clinic was significantly impacted by the COVID-19 pandemic with greater than 10% of clinic patients dying from complications of SARS-CoV-2 infections in March and April 2020, thus studying the effectiveness of the vaccine in our center was research priority.” We have also added the following to the results, “Of the one hundred and forty patients approached in this Middle Atlantic urban hemodialysis facility, 30 patients volunteered to participate and to have their antibody levels checked monthly from March 2021 through March 2022.”

Admittedly, our study's sample size is constrained. However, we assert that our research carries certain unique features that augment its worth. Firstly, the primary composition of our patient population is African American (97%). Although this attribute might ostensibly restrict the generalizability of our findings, it concurrently provides a distinctive viewpoint, particularly in light of the paucity of research investigating vaccine efficacy and persistence amongst African American patients undergoing hemodialysis. Secondly, our findings underscore the potential significance of post-vaccination antibody testing strategies, thereby furnishing invaluable insights that could aid in the optimization of vaccination strategies.

  1. Also, 97% of the participants were African Americans. The potential impact of this on the results is not discussed.

Response: In response, we have expanded our discussion to address your concerns, for example: “The mRNA-based COVID-19 vaccines have demonstrated substantial efficacy across diverse ethnic groups, and previous studies suggested an elevated vaccine efficacy in the African American population. Our cohort, predominantly composed of African American participants, manifested an antibody positivity rate that trended towards the higher end of previously recorded seropositivity rates, aligning with existing literature.”

  1. The authors repeatedly use the term “COVID-19 infection”. This is inappropriate since COVID-19 is the disease itself, caused by an infection by Sars-CoV-2. Similarly, one wouldn't say "AIDS infection". It is either AIDS, the disease, or an infection caused by HIV, which aren't necessarily the same.

Response: In our resubmitted manuscript, we have corrected the “COVID-19 infection” into “SARS-CoV-2 infection”.

  1. Technically, there were two further subgroups, one receiving the Pfizer, and the other one receiving the BioNTech vaccine. We cannot presume that they had the exact same effect.

Response: We concur with your viewpoint that BNT162b2mRNA (Pfizer) and mRNA-1273 (Moderna) vaccines may prompt distinct immunological reactions. Within our study population, only two patients were administered the mRNA-1273 vaccine series, with one from the COVID-19 naïve group and the other from the recovered group, as detailed in Table 1. The limited sample size hindered our ability to discern any significant disparities in vaccine outcomes between Pfizer and Moderna. However, we have incorporated a reference into our discussion to highlight the potential variances between these two vaccines. This citation suggests that durable antibody responses are closely related to higher initial antibody titers, as observed in peritoneal dialysis patients and those who received the mRNA-1273 vaccine.

  1. How was the arbitrary cutoff value for "responders" and "nonresponders" determined? I see no reference for this.

Response: We have rewritten this part for improved clarity in the methods-explanatory variables section. According to the instructions provided with the Beckman Coulter Access SARS-CoV-2 IgG Antibody Test Kit, a seropositive response was defined as an antibody titer level equal to or above 1.00 S/CO SARS-CoV-2 IgG. Any value below this threshold was considered seronegative. In our study, patients who remained seronegative following two vaccine doses were defined as non-responders. Conversely, those who exhibited a seropositive response after the administration of two doses were defined as responders.

Once again, we appreciate your insightful comments and suggestions. We would be glad to respond to any further questions and comments that you may have. We hope that the revised manuscript meets your expectations and could be considered acceptable for publication.

Best regards,

Mingyue He, Avrum Gillespie

Reviewer 2 Report

Please see the comments bellow.

1.       Incorporated the objective in abstract

2.    Peak antibody levels post-booster 27 was similar in both groups (27.1 ± 3.9 vs 37.9 ± 8.2, p = 0.20). This is not similar so correct it i.e- increased but not significant or both grouped showed not significant difference.

3.       Recovered should be either small or capital letter. Correct it in whole manuscript.

4.       They received same vaccine in next dose or different from the previous.

5.     This APR increase and decrease based on subjects but it is not clear this happened to same or different subjects. Provide the antibody titer level at each dose. Otherwise, it is not clear, what is the meaning of decrease and increase after dosing. What is the time interval between the dose?

6.       What is the peak time for antibody after 1st and 3rd dose? It was done in the same subjects or different.

7.     From which level to ..? just saying 12 or 20 %, does not make sense so please added the initial antibody level.

8.       What is the 863 in discussion? 

Authors need to improve their writing because it has a lot of grammar mistakes.

Author Response

Dear Reviewer:

On behalf of all the contributing authors, we sincerely appreciate the opportunity to resubmit the revised draft of our manuscript. Your invaluable time and effort, along with the detailed feedback and constructive comments, have significantly contributed to enhancing the quality of our work. We are presenting our responses to the reviewers' comments and concerns in a point-by-point manner.

Reviewer comments are presented below in italicized font, with individual concerns numbered for easy reference. Our responses follow in regular font. Changes and additions made to the manuscript based on these suggestions are indicated in red text.

Comments from Reviewer:

  1. Incorporated the objective in abstract.

Response: Thank you for your feedback. We have incorporated the objective into the abstract as follows: “To guide vaccination strategies in patients on dialysis, it is critical to characterize the longevity and efficacy of the vaccine.”

  1. Peak antibody levels post-booster 27 was similar in both groups (27.1 ± 3.9 vs 37.9 ± 8.2, p = 0.20). This is not similar so correct it i.e- increased but not significant or both grouped showed not significant difference.

Response: We appreciate your observation and advice on this point. In line with the suggestion, we have changed the expression to “Peak antibody levels post-booster showed no significant difference between both groups (27.1 ± 3.9 vs 37.9 ± 8.2, p = 0.20).”

  1. Recovered should be either small or capital letter. Correct it in whole manuscript.

Response:  Based on your comments, we have made corrections to ensure the word “recovered” is used consistently throughout the entire manuscript.  

  1. They received same vaccine in next dose or different from the previous.

Response: We have rewritten the “Vaccination Status” section for clarity in the revised manuscript as follows: “Patients who initially received the BNT162b2 mRNA vaccine were subsequently administered the BNT162b2 mRNA vaccine for their second and third doses. Similarly, those who were initially administered the mRNA-1273 vaccine received the mRNA-1273 vaccine for their second and third doses.”

  1. This APR increase and decrease based on subjects but it is not clear this happened to same or different subjects. Provide the antibody titer level at each dose. Otherwise, it is not clear, what is the meaning of decrease and increase after dosing. What is the time interval between the dose?

Response: We apologize for any confusion caused by the original text. We have made the following modifications to the original manuscript: “The Antibody Positivity Rate (APR) is defined as the percentage of individuals who show a seropositive response out of the total number of patients interested. Among the 22 COVID-19 naive patients in our study, 12 exhibited seropositivity after the first vaccine dose, resulting in an APR of 55% (12/22). This rate increased to 91% (20 out of 22) after the second dose.”; “Of the COVID-19 naïve patients, 86% (19/22) received a third booster dose, and the APR for patients who received the third dose was 100% (19/19).”

The antibody levels following each dose are detailed in the main text and are also summarized visually in Figure 4.

Regarding the time interval between the doses: “The average interval between the first and second doses was 32 ±5 days, while the mean period between the second dose and the third booster doses was 6 ± 0.4 months. There were no significant differences in dosing intervals between the COVID-19 naive group and the COVID-19 recovered group (Table 1).”

  1. What is the peak time for antibody after 1st and 3rd dose? It was done in the same subjects or different.

Response: Given that the antibody levels were tested monthly and the average interval between the first and second dose was approximately 32 days, most of patients only had one antibody level reading available after the first dose. This reading was lower than the antibody levels recorded after the second dose. Consequently, the antibody response exhibited two peaks following the three doses: one after the second dose, and the other after the third dose. In our study, all thirty participants received two doses of mRNA vaccines. Most participants (26 out of 30) received a third booster dose. The four patients who did not receive the third dose were excluded from analyses involving post-third dose data. “Antibody levels of all participants peaked at 60 ± 6 days post-second dose. The duration to reach peak levels after the second dose showed no disparity between the COVID-19 naïve and recovered Groups (65 ± 8 days vs 45 ± 3 days, p = 0.15).” “Among participants who received the third dose, antibody levels peaked at 38 ± 4 days post-third dose. There was no significant difference in the time interval to reach peak levels after the third dose between the COVID-19 naïve and the recovered Group (36 ± 4 days vs 42 ± 13 days respectively, p = 0.55).”

  1. From which level to ..? just saying 12 or 20 %, does not make sense so please added the initial antibody level.

Response: We have rewritten this part according to your suggestion. “Both the COVID-19 naïve Group and the COVID-19 recovered Group demonstrated a significant decline in antibody response when comparing the peak antibody levels post-second dose and the levels before the third booster dose. For patients who did not receive the third dose, the antibody level recorded around six months post-second dose was utilized for analysis. Of the 22 COVID-19 naive patients, the mean antibody level significantly decreased from 17.9 ± 3.2 post-second dose to 5.6 ± 2.4 pre-third dose (delta 12.3 ± 2.3, p < 0.001 by paired t-test). Among the 8 COVID-19 recovered patients, the mean antibody level significantly decreased from 44.7 ± 5.6 post-second dose to 22.7 ± 5.2 pre-third dose (delta 22.0 ± 2.7, p < 0.001 by paired t-test).”

  1. What is the 863 in discussion?

Response: We apologize for any confusion that might have arisen due to the inclusion of the word count for the discussion section (863 words). For clarity and to maintain the conciseness of the manuscript, we have now removed it.

  1. Authors need to improve their writing because it has a lot of grammar mistakes.

Response: Thanks for your suggestion. We have carefully reviewed the paper.

Once again, we appreciate your insightful comments and suggestions. We would be glad to respond to any further questions and comments that you may have. We hope that the revised manuscript meets your expectations and could be considered acceptable for publication.

Best regards,

Mingyue He, Avrum Gillespie

Round 2

Reviewer 1 Report

The authors addressed all issues raised by this reviewer. I have no further concerns regarding this manuscript.

The English of the study is appropriate and easy to follow.